# The Turkish Cypriot Municipalities' Productivity and Performance: An Application of Data Envelopment Analysis and the Tobit Model

**Dilber Çağlar Onbaşıoğlu**

Department of Accounting, Karmi Campus, Girne American University, University Drive,
P.O. Box 5, Karaoglanoglu, Kyrenia 99428, Turkey; caglardilber@hotmail.com

**Abstract:** Nowadays, countries are more concerned with the improvement of effectiveness and efficiency in public sector activities in the perspective of frugal innovation. The problem centers around how to obtain more and better public service with the limitations of the public incomes and indebtedness in preserving environmental conditions. This paper empirically investigates the efficiency, technical efficiency, productivity, and the determinants factor of implementing sustainable development policy of the five major municipalities in North Cyprus by conducting DEA and Tobit analyses during the period from 2004 to 2018 quarterly. The empirical results show that the size of the economically active population of a city, lower expenditures, and grants result in a higher efficiency, whereas the independent revenue sources (grants) and the per capita expenditures of North Cypriot municipalities have a negative effect on the efficiency. The employment rate in the municipalities has a considerable negative effect on the efficiency score. The results of Tobit analysis also show that population has a positive impact which may increase the technical efficiency. Finally, the findings of this study demonstrated that implementing proper environmental programs not only improve the efficiency of local government but also help the ecological sustainability and the geographical location of the regional changes and barriers for sustainable initiatives by using proper waste mechanism, clean water technology, and solar lighting.

**Keywords:** municipalities; DEA; Tobit analysis; sustainable development; efficiency; employment; ecological sustainability; frugal innovations



## 1. Introduction

The research on the efficiency and productivity of local governments has important practical significance on realizing frugal and intensive utilization of services and improving the efficiency of services used to promote the sustainable development of economic and social. Local governments have a very important role in stimulating sustainable development. The Sustainable Development Goals, which were firstly introduced by United Nations, tried to establish a relationship between the economic, social, and ecological dimensions and provide a way for local communities to be involved in the development processes with a need for sustainable consumption and production. It is very important for local governments to provide services in a frugal, efficient, and effective manner in increasing the welfare level of the society and increasing the quality of life. Most local governments are concentrating on administrative reforms to improve governmental practices and operations in the most efficient and effective way. The objective of this study is to investigate the levels of both efficiency and service quality between five major municipalities in North Cyprus and to determine that there are similarities between each municipality with respect to their financial, socio-economic, and budgetary characteristics as well as frugal innovation by providing proper waste mechanism, clean water technology, and solar system for lighting. The first municipality is Nicosia, which is the capital and the largest city of the Turkish

Republic of Northern Cyprus. Then, Famagusta, Kyrenia, Guzelyurt, and Iskele, which are the other major municipalities.

Although several researchers have been studying the performance of local governments in other developed countries by applying the Data Envelopment Analysis (DEA) method, this study is going to be the first to study local governments in North Cyprus. Most studies on local governments' performance come from European countries, including Gonzalez et al. (2011); Benito et al. (2010); Geys and Moesen (2009); Worthiington and Dollery (2002); Gimenez and Prior (2007); Kutlar et al. (2012); Fogarty and Mugera (2013); Storto (2016); and Narbón-Perpiñá and Witte (2018). There are some recent studies related to the other countries (Sihaloho 2019; Tu et al. 2017). In addition, Narbón-Perpiñá and Witte (2018) prepare a systematic review of the literature on the public sector efficiency by comparing their data and samples, and using techniques for measuring efficiency and summarizing the inputs and outputs that are employed. Their results obtained from 84 empirical studies firstly suggests that most of the previous studies have focused only one of the following approaches: DEA, FDH, or SFA, by considering cross-sectional data. Then, the best-studied countries on this topic were accepted, including Spain, which is the most analyzed country with 13 papers; Belgium, with 9 papers; and finally, Germany, with 8 papers. In addition, their review study shows that the determination of inputs and outputs is a complex task and leads to a difficulties in to collect and measure the data.

The Table 1 summarizes the most popular studies on local government especially the studies based on DEA, SFA, or VECM techniques that are the most preferable techniques to test the fiscal conditions of municipalities.

**Table 1.** Previous studies on local governments.

| Author | Municipality | Method | Results |
|---|---|---|---|
| Deller et al. (1988) | 1799 Illinois, Minnesota, Ohio and Wisconsin municipalities (1982) | Cost Function | Joint use of inputs produces lower overall costs, emphasis on local government consolidation or contract tendering to capture scale. |
| Grosskopf and Yaisawarng (1990) | 49 Californian municipalities (1982) | DEA | Economies of scope that requires specialization or diversification. |
| Hayes and Chang (1990) | 191 US municipalities (1982) | Stochastic Frontier | No differences has found between the different municipal government structures. |
| Deller and Nelson (1991) | 446 Illinois, Minnesota and Wisconsin municipalities (1990) | DEA | Increases in jurisdictional size related to improvement in efficiency. |
| Deller (1992) | 1319 Illinois, Minnesota and Wisconsin municipal | Stochastic Frontier | Local government need for productive efficiency due to federal policy and fiscal pressures. |
| Deller et al. (1992) | 435 Illinois, Minnesota and Wisconsin municipal area (1982) | Stochastic Frontier | High cost inefficiency can be decreased by input consolidation. |
| De Borger et al. (1994) | 589 Belgian Municipalities (1985) | FDH | The main determinants of municipal efficiency are scale and fiscal revenue capacity. |
| Deller and Halstead (1994) | 104 Maine, New Hampshire and Vermont Municipalities (1987) | Stochastic Frontier | It is highly inefficient to place a rural road maintenance and positive outcomes from higher levels of training. |
| De Borger et al. (1994) | 589 Belgium Municipalities (Horizontal section) | FDH | Mean efficiency scores range from 0.57 (COLS) to 0.94 (FDH). |
| De Borger and Kerstens (1996) | 589 Belgium Municipalities (Horizontal section) | DEA and FDH Stochastic method | Mean efficiency scores range from 0.86 to 0.95 depending on specification. |

**Table 1.** *Cont.*

| Author | Municipality | Method | Results |
|---|---|---|---|
| de Sousa and Ramos (1999) | 701 Brazil municipalities (Horizontal section) | DEA and FDH | Smaller municipalities are less efficient than bigger municipalities. |
| Worthington and Dollery (2000) | 166 Australia municipalities (Horizontal section) | DEA and Stochastic method | Mean efficiency scores range from 0.70 (DEA) to 0.87 (SFA). With 70% decrease in inputs the municipalities could become efficient. |
| Prieto and Zofio (2001) | 209 Spain municipalities whose population is under 20.000 (horizontal section) | DEA | Mean efficiency score range from 0.60 (DEA) to 0.85 (SFA). |
| Balaguer-Coll et al. (2002) | 258 Spain municipalities (Panel data) | DEA | Bigger population and level of traditional activity have positive effect on the efficiency but high tax person, high scholarship per person have negative effect on the efficiency. |
| Loikkanen (2006) | 353 Finnish municipalities (panel data) | DEA | Averages of the annual median efficiency scores range from 0.86 to 0.90 depending on the specification used. |
| Afonso and Fernandes (2006) | 51 Lisbon region municipalities | DEA | Mean efficiency scores range from 0.33 to 0.73 depending on the specification used |
| Geys (2006) | 304 Flemish local governments in 2000 | SFA | Output can on average e increased 14% compared to most efficient. |
| Tanaka (2006) | 3017 Japanese municipalities in Kinki area in 2001 | SFA | Inputs could be reduced by about 12% on average. |
| Sung (2007) | 222 Korean Local Governments from 1999 to 2001 | DEA | Mean efficiency scores range from 0.57 to 0.99 depending on the specification used. |
| Balaguer-Coll et al. (2007) | 414 Spanish local governments (located in Valencia) in 1995 | DEA, FDH | Mean efficiency scores range from 0.53 to 0.90 depending on the specification used. |
| Borge et al. (2008) | 362–384 Norwegian municipalities from 2001 to 2005 | Ration | Average output 35% below most efficient. |
| Geys and Moesen (2009) | 304 Flemish local governments in 2000 | DEA, FDH, SFA | Mean efficiency scores range from 0.50 (DEA) to 0.95 (FDH) and 0.86 (SFA). |
| Balaguer-Coll and Prior (2009) | 258 Spanish municipalities (1992–1995) | DEA | The local government efficiency is determined by the size of the municipality, the per capita tax revenue, the per capita grants and the amount of commercial activity Mean efficiency scores range from 0.62 to 0.76 depending on the specification used. |
| Benito et al. (2010) | 31 Spanish local governments in 2002 | DEA | Mean efficiency scores range from 0.32 to 0.84 depending on the area of public good provision. |
| Sole-Olle and Sorribas-Navarro (2011) | Spanish municipalities (1988–2006) | Vector error correction model (VECM) | Grants have accepted the most important role in the adjustment process in environment and it results in a moral hazard problem. The results also refers that the viability of the local finance system is feasible with different institutional arrangements. |
| Kutlar et al. (2012) | 27 Turkish municipalities (2006–2008) | DEA and Malmquist index | There was a decrease in the number of efficient municipalities and the level of their efficiencies since 2006. |

**Table 1.** *Cont.*

| Author | Municipality | Method | Results |
|---|---|---|---|
| Fogarty and Mugera (2013) | 98 Local Councils in Western Australia (2009–2010) | DEA | Some of the councils were operating under increasing returns to scale and some of them were operating under decreasing returns to scale. In order to achieve an optimal scale for local councils there is room for either scaling down or expanding. |
| Storto (2016) | 108 Italian municipalities | DEA | Scale inefficiencies were found in a number of municipalities and there was a trade-off between expenditure efficiency and effectiveness. |
| Cárcaba et al. (2017) | Spanish Municipalities (2001–2011) | DEA and Malmquist index | Positive social progress with an average improvement of about 5% during the decade and positive catching up is measured in all regions. |
| Sihaloho (2019) | Local government expenditure in regencies and cities in Jest Java (2001–2010) | DEA and Tobit | Many regions have high spending but cannot achieve the maximum score of efficiency. In addition according to the Tobit results optimal technical efficiency scores is positively affected by investment credit funding and total labor. |
| Aiello and Bonanno (2019) | Review study on Local Government efficiency | Meta Regression | The research does a meta-analysis of 360 observations gleaned from 54 papers published between 1993 and 2016. The findings reveal that the efficiency scores belongs to the studies that focus on technical efficiency is higher than studies that evaluate cost efficiency, using panel data rather than cross-section data also improves efficiency. Surprisingly, research that utilize the FDH method produce greater efficiency scores on average than studies that use the DEA method. |
| Narbón-Perpiñá et al. (2019) | Spanish local governments | four different non-parametric methodologies | Examine overall cost efficiency throughout the economic downturn (2008–2013). The findings imply that local government efficiency in Spain improved between 2008 and 2013, since budget expenditures (inputs) decreased but local public services and facilities (outputs) remained constant. |
| Olejniczak (2019) | local governments in Poland | DEA | Investigate the possibility of a link between a municipality's economic potential and the efficiency (relative) of its operations. The findings indicate that there is a link between the commune's revenue and its effectiveness in operation. |
| Benito et al. (2019) | Spanish municipalities | DEA | It examines the effectiveness of small municipalities' drinking water supply services and the findings reveal that population density and residents' income levels have a negative and considerable impact on the efficiency of drinking water delivery. When the provision of drinking water is administered directly by the local government, it is more efficient. |

**Table 1.** *Cont.*

| Author | Municipality | Method | Results |
|---|---|---|---|
| Rambe (2020) | Local governments in North Sumatra Province | DEA | Determine the relative efficiency of educational spending in terms of completing the years of schooling. They concluded that the average degree of relative effectiveness of 33 local governments in North Sumatra dropped from 2015 to 2018 and a regional division has failed to improve the relative efficiency of local governments. |
| Lee et al. (2020) | Korea's R&D investment performance on 16 local governments | DEA | Concluded that "R&D investment efficiency into pure R&D investment technical efficiency and scale efficiency and derived implications regarding the input scales". |
| Tran and Dollery (2020) | The Victorian local government system | DEA | Over the years 2014–2015 to 2017–2018, look into the relationship between operational efficiency and local resident satisfaction for three main municipal types; rural, regional and metropolitan. Although there is a substantial link between efficiency and satisfaction for metropolitan and regional councils, the same is not true for rural councils, according to the findings. |
| Pougkakioti and Tsamadias (2020) | Municipalities in Greece | DEA an Malmquisitc Index | This study looks at how municipalities' relative efficiency and productivity changed from 2013 to 2016. According to the empirical evidence, efficiency and productivity have gradually improved following the most recent Local Government reform and under tight budgetary policy. |
| D'Inverno et al. (2020) | Flemish Municipalities | DEA | Analyzes the relationship between the size of a municipality and the availability of local services. The major findings point to scale inefficiencies and give only SMALL evidence for a 10,000-person optimal size of local public good provision. |
| Plaček et al. (2020) | Municipalities in the Czech Republic. | DEA and FDH | The impact of policies that promote excellence on actual performance is examined in this article. The difference impact approach was used to analyze the results, and it was discovered that this particular public policy had no positive impact on municipal efficiency. The reverse is obtained when the difference-in-differences approach is utilized. |
| Ziemba (2021) | Local Governments in Poland | Survey Data | ICT quality, information culture, and ICT management have a significant positive impact on the sustainability of local governments |
| Tran and Dollery (2021) | Local Governments in South Australia | DEA | The technical efficiency of local government is investigated in this study by looking input excess in various municipal services. Using the bootstrapping method to filter out the impact of environmental factors on input abundances and unobserved disturbances, the results show that residual inefficiency is due to management inefficiency. |

As is summarized in the above table, four different approaches have been applied on local public sector efficiency; (i) the deterministic frontier approach (DFA); (ii) the stochastic frontier approach (SFA); (iii) the data envelopment analysis (DEA) and the Free Disposal Hull or FDH approaches. DFA is an econometric approach which argues that all deviations from the frontier are the result of inefficiency. The SFA is an econometric technique that assumes two-component error structures, therefore the random error s is normally distributed and the inefficiencies usually follow an asymmetric half-normal distribution. The stochastic frontier analysis (SFA) method was developed by Aigner et al. (1977), Battese and Broca (1997), Battese and Corra (1977); Jondrow et al. (1982), and Battese and Coelli (1988) to estimate the efficiency in production by introducing a two-part error term in a regression mode. One is to measure error by applying ordinary stochastically noise and the second focusses on the inefficiency by applying a disturbance term. The DEA is a mathematical programming technique that assumes all deviations from the estimated frontier represent inefficiency. The FDH is a variant of DEA and considers the assumption that the production technology to be kept to a minimum.

This study attempts to contribute to existing literature in two ways. Firstly, there is extensive literature regarding the modelling and empirical investigation on the local governments by applying similar techniques, such as DEA or VECM, but this study uses all popular techniques plus the Tobit analysis. Secondly, there are some analyses related to developed and developing countries, but this study is the first in North Cyprus. Therefore, this study empirically applies to test the efficiency of five major municipalities in North Cyprus and the results may be considered by local governments on the implementation of a sustainable development policy for their work in small island countries. The study examines the relationship between the environmental management and financial performance. Therefore, this study is of great importance because it is applied for the first time in Northern Cyprus, and no studies have been conducted to measure the financial situation and efficiency of municipalities until today. The results obtained from this study aim to shed light on the more efficient management of district municipalities for government officials.

## 2. Materials and Methods

Economic efficiency refers to the jurisdictions providing the maximum amount of output for given levels of input or the required minimum level of inputs for a give level of output. Efficiency measurements can be performed in two steps. The first step is based on determining the best combinations of inputs and outputs that designate optimal or efficient behavior. Then, in the second step, the level of efficiency or inefficiency is determined by comparing each jurisdiction with the best performing jurisdiction. While the value of input or output decisions are hard to resolve, the literature refers to different types of techniques to measure the efficiency in terms of technical, economics, parametric, and nonparametric methods. Nowadays, DEA is one of the most popular nonparametric analyses to determine the efficiency of health, education, finance, production, and services of the public sector. In the literature, most of the studies related to the efficiency of the public sector were tested by DEA analysis. DEA is a mathematical programming approach that was developed by Charnes et al. (1978) to estimate an empirical production frontier by applying input/output data and measure the relative efficiency. The technique is based on the production possibilities to obtain an empirical frontier and measure efficiency as the distance to the frontier. The variables used as inputs are the factors that have a cost and should be kept at a minimum, and the outputs are the products that have a positive value and should be increased and maintained at a maximum. The weight of these inputs and outputs are used to obtain a precise index of productive efficiency, focus on the decision making unit (DMU), and maximize (minimize) the weighted output/input ratio of each decision making unit (DMU). Then, one must measure the distance from the linear frontier to the DMU under evaluation. In addition, each DMU is assigned an efficiency score which depicts the distance range between the values of 0 and 1. DEA is a non-parametric frontier analysis method that is used to measure the efficiency of production in firms and public

organizations. This study is based on two methods; first, the data envelopment analysis (DEA) is applied to measure the problems of efficiency and the mathematical formulation method which is usually to evaluate efficiency by inputs as follows;

$$
\begin{aligned}
&\text{Min}\theta, \lambda, \theta \\
&\text{s.a.} - yi + Y\lambda \geq 0 \\
&\theta xi - X\lambda \geq 0 \\
&N1'\lambda = 1 \\
&\lambda \geq 0
\end{aligned}
\tag{1}
$$

where θ is the coefficient that represents the proportionally reduced inputs of the evaluated unit and its value measures the efficiency of unit 'i' subject to evaluation (DMU). X represents input matrices and Y represents output matrices, whereas, xi and yi are defined as the observed inputs and outputs corresponding to the DMU under evaluation, and λ is the active vector that is used for comparison to determine unit 'i'. It is a type of method that compares each producer with the best producer to determine the best frontier.

The model occurs with two restrictions; the first restriction (Yλ ≥ yi) enables us to determine as many outputs as those obtained by the DMU under analysis, and the second restriction (θxi ≥ Xλ) results in determining the lowest possible input consumption. So, by testing each unit, we will obtain a coefficient of θ for each DMU. According to the results, the DMU will be defined as efficient if θ = 1, otherwise it is defined as inefficient. It is based on constant returns to scale. It is not similar to the normal statistical approaches evaluations that evaluate the units according to the averages. DEA is often used to evaluate the efficiency of a number of producers by comparing each one of them only with the best producers.

The measurement is usually applied as follows;

$$
\text{Efficiency} = \frac{\text{Weighted sum of outputs}}{\text{Weighted sum of inputs}}
\tag{2}
$$

In the literature, many works have analyzed other sectors, such as education and health, but only a few works have analyzed the evaluation of municipal services. The main reason for this can be accepted as the availability of data. Obtaining data from municipalities is not as easy as the other sectors, and even if data are available, the researchers always face problems of measuring public outputs. With limited and restricted data for local governments in North Cyprus, DEA can be accepted as the best measure for our analysis, because one of the important advantages of DEA is that it works well with small samples. The popularity of the DEA model comes from some advantages. For example, in order to determine the most efficient decision making units (DMUs), it does not require a particular functional form, or the assumption to be made about the distribution of inefficiency. In addition, DEA does not require the assumption about the form of production function, and the relationship between the variables can be evaluated by bivariate statistical tests. DEA is also able to handle a high number of variables and relationships. It can handle multiple inputs and outputs in the production process of the local governments.

On the other hand, one of the important disadvantages of DEA is its low discriminating power. This especially occurs if the sample size is limited and when many dimensions are taken into account. Some studies apply value efficiency analysis (VEA) to improve both the discriminating power of DEA and the consistency of the weights on which the evaluation is based on. In order to apply VEA, the first DEA frontier must be applied.

In the non-parametric approach, there are three alternatives to measure productivity changes. These are Fischer index (1992), Tornquist index (1936), and Malmquist index (1953). We use a Malmquist index methodology which examines the changes in productivity to provide information on technical efficiency and technological change. Technical efficiency measurements consider how well the inputs are converted into outputs through the production process. Malmquist index is a test to measure both production and technical

efficiency separately. Malmquist index was used by Caves et al. (1982) in DEA analysis. It is a type of index that can be calculated with parametric and linear programming methods. This index is based on the amount of input and output and determined by distinction functions that represents multiple inputs and multiple outputs technologies. It is suitable for a type of sector where the prices are not determined exactly, such as the public sector. Because of these reasons, the Malmquist index prices and assumptions, as well as the structure of the technology, are not needed, which makes it superior to the other indexes.

Then, the Tobit model is used to estimate linear relationships between variables:

$$
\begin{aligned}
y_i^* &= x_i'\beta + \mu_i \quad (i = 1, 2, \ldots, n) \\
y_i^* &> 0 \; if \; y_i = y_i^* \\
y_i^* &\leq 0 \; if \; y_i = 0
\end{aligned} \tag{3}
$$

$x_i'$ = independent variable
$y_i$ = dependent variable; 0 or 1
$\beta$ = estimated coefficients
$u_i$ = error term.

*Sample and Data*

The data sources are obtained from the Ministry of Internal Affairs (2018). The data variables are chosen based on the research by De Borger et al. (1994), Bosch et al. (2000), and Balaguer-Coll and Prior (2009).

The Table 2 presents the output indicators based on the minimum services provided (source: Balaguer-Coll et al. (2007)).

**Table 2.** Output indicators based on the minimum services required.

| Minimum Service Provided | Output Indicators |
|---|---|
| Public street lighting | Number of lighting points |
| Cemetery | Total population |
| Waste collection | Waste Collected |
| Street cleaning | Street infrastructure surface area |
| Supply of drinking water to households | Population, street infrastructure surface area |
| Access to population centers | Street infrastructure surface area |
| Surfacing of public roads | Street infrastructure surface area |
| Regulation of food and drink | Total population |

Then the total number of inputs and outputs that are applied to measure local government performance is summarized in Table 3.

**Table 3.** Input and output Variables for Measuring Performance.

| Input Variables | Output Variables |
|---|---|
| $X_1$ = Wages and salaries expense | $y_1$ = Population |
| $X_2$ = Operating expenditure | $y_2$ = Number of lighting points |
| $X_3$ = Current and capital transfers | $y_3$ = number of waste collected |
| $X_4$ = capital expenditure | $y_4$ = homes with clean water |

Under the aim to determine what essentials services are, descriptive statistics corresponding to those variables are estimated, and the results are summarized in Table 4.

**Table 4.** Descriptive Values of the Variables.

| Variable | Mean | SD | Min | Max |
|---|---|---|---|---|
| Population | 26,591 | 16,477.4 | 3511 | 56,146 |
| NOLP | 44.98 | 36.17 | 4 | 126 |
| NOWC | 52.35 | 31.10 | 5 | 100 |
| HWCW | 226.13 | 94.27 | 36 | 399 |

Then, before we do the regression model, the multicollinearity is also tested, and the linear degree is determined by focusing on Pearson's linear correlation coefficient. After that, with an estimation of the Cobb-Douglas cost function (see Table 5), we determine that the behavior of local governments with the DEA model will depend upon the corresponding variables specification.

**Table 5.** Results of the Cobb-Douglas cost function.

| Model | Independent Variables | B (t Student) |
|---|---|---|
| 1 | Population | 0.52 (8.59) ** |
| 2 | NOLP | 0.53 (2.01) * |
| 3 | NOWC | 0.21 (1.39) |
| 4 | HWDW | 0.23 (2.71) * |

Notes: Model 1: $R^2$ adjusted = 0.65, Model 2: $R^2$ adjusted = 0.44, Model 3: $R^2$ adjusted = 0.48, Model 4: $R^2$ adjusted = 0.49. * and ** denotes statistical significance at 0.01 level and 0.05 level respectively.

In the regression models, the explanatory is good, especially in model 1, since adjusted R2 represent a good value. According to the regression results in each model, we can say that the population includes very good explanatory variables since adjusted R2 in model 1 is better that the other models.

The total expenses, (x1) wages and salaries expense, (x2) operating expenditure, (x3) current and capital transfers, and (x4) capital expenditure are considered as input variables. The output variables are: (y1) population, (y2) number of lighting points, (y3) number of waste collected, and (y4) number of homes with clean water.

In the output determination, the variables are represented as the most important services that are provided by local governments. These input and output variables are the most important factors for the efficiency of local governments and they should be taken into account together with environmental management practices. Therefore, when we analyze these inputs and outputs, we also measure the production of local governments on the planning activities, responsibilities, practices, and resources that they deal with to improve, provide, and maintain the environment with ecological sustainability. The Table 6 shows the outputs and inputs used in this study.

One of the benefits of this study is that the approach utilized (DEA) has been used in many comparable studies before, enlightening our study. Furthermore, data collecting is one of the most difficult aspects of the study. Due to a lack of appropriate data, the study did not include data from the previous years, and only five large district municipalities were included. In addition, other exogenous factors may affect municipal efficiency. The method used in this study is inadequate to measure and distinguish efficiency changes caused by these exogenous causes.

**Table 6.** Definition of variables.

| Variables | Definitions |
|---|---|
| Wages and salaries expense (WSEXP) | Total expenses for salary and social security expenses |
| Operating expenditures (OEXP) | Total expenses to provide services (such as purchases of goods and services |
| Current and capital transfers (TREXP) | Total transfers for financially needed people (such as transfer to non-profit organizations and households) |
| Capital expenses (CAPEXP) | Total expenses for capital (purchased of manufactured goods, cost of producing moveable and immovable goods) |
| Population (POP) | Number of people belongs to each municipalities (population of municipality according to 2011 census) |
| Number of lighting points (NOLP) | NOLP = Amount of revenue obtain from lighting/number of houses exist in the municipality district |
| Number of waste collected (NOWC) | NOWC = Total revenue from waste collections/number of houses exist in the municipality district |
| Homes with clean water (HWCW) | HWCW = Total revenue from drinking water/number of houses exist in the municipality district |
| Technical Efficiency Scores (TEF) | Calculation results from cost efficiency analysis in DEA |
| Scale Efficiency Scores (SEF) | Calculation results from scale efficiency in DEA |
| Malmquist scores (MALEF) | Calculation results from Malmquist index |

## 3. Results and Discussion

This part focuses on the measurement and evaluation of performance in North Cyprus local governments by applying data envelopment analysis (DEA) and Malmquist index. The technical efficiency and productivity of the municipalities based on input and output data were measured and evaluated with DEA. There are no differences in the structures, tax system, or functions between municipalities in North Cyprus.

Table 7 shows the technical efficiency scores, inefficiency scores, and the technical efficient ranking for DEA test results of five municipalities (Nicosia, Famagusta, Kyrenia Guzelyurt, and Iskele) over the period 2004 to 2018. In terms of technical efficiency, the average technical efficiency score of the five major municipalities in North Cyprus is 0.66 (see Table 9). According to the results that are summarized in Table 7, the Famagusta municipality is accepted as the most efficient, and Iskele is the most inefficient in terms of technical efficiency when compared to the other four municipalities.

Table 8 represents the technical efficiency scores and productivity scores based on the Malmquist index. The Malmquist index results exhibit that the average technical efficiency score of North Cypriot municipalities between the years 2004 and 2018 is 0.77 (see Table 9). According to the Malmquist results. Famagusta has the highest productivity scores. One of the reasons why Famagusta has become the most efficient municipality can be considered as the geographical location of the regions changes barriers for sustainable initiatives.

**Table 7.** DEA Technical Efficiency Scores.

| | Input Items: (1) WSEXP, (2) OEXP, (3) TREXP, (4) CAPEXP Output Items: (1) POP, (2) NOLP, (3) NOWC, (4) HWCW | | | |
|---|---|---|---|---|
| Year | DMUs | Technical Efficiency Scores | Technical Rank | Technical Inefficiency Sores |
| 2004 | Lefkosa | 0.83 | 4 | 0.17 |
| 2004 | Famagusta | 0.86 | 1 | 0.14 |
| 2004 | Kyrenia | 0.85 | 2 | 0.15 |
| 2004 | Guzelyurt | 0.84 | 3 | 0.16 |
| 2004 | Iskele | 0.45 | 5 | 0.55 |
| 2005 | Lefkosa | 0.82 | 3 | 0.18 |
| 2005 | Famagusta | 0.85 | 1 | 0.15 |
| 2005 | Kyrenia | 0.84 | 2 | 0.16 |
| 2005 | Guzelyurt | 0.85 | 1 | 0.15 |
| 2005 | Iskele | 0.45 | 4 | 0.55 |
| 2006 | Lefkosa | 0.57 | 3 | 0.43 |
| 2006 | Famagusta | 1 | 1 | 0 |
| 2006 | Kyrenia | 0.78 | 2 | 0.22 |
| 2006 | Guzelyurt | 0.53 | 4 | 0.47 |
| 2006 | Iskele | 0.30 | 5 | 0.70 |
| 2007 | Lefkosa | 0.65 | 4 | 0.35 |
| 2007 | Famagusta | 0.88 | 2 | 0.12 |
| 2007 | Kyrenia | 0.97 | 1 | 0.03 |
| 2007 | Guzelyurt | 0.76 | 3 | 0.24 |
| 2007 | Iskele | 0.32 | 5 | 0.68 |
| 2008 | Lefkosa | 0.36 | 3 | 0.64 |
| 2008 | Famagusta | 0.60 | 1 | 0.40 |
| 2008 | Kyrenia | 0.30 | 4 | 0.70 |
| 2008 | Guzelyurt | 0.40 | 2 | 0.60 |
| 2008 | Iskele | 0.60 | 1 | 0.40 |
| 2009 | Lefkosa | 0.64 | 3 | 0.36 |
| 2009 | Famagusta | 0.70 | 1 | 0.30 |
| 2009 | Kyrenia | 0.65 | 2 | 0.25 |
| 2009 | Guzelyurt | 0.42 | 5 | 0.58 |
| 2009 | Iskele | 0.60 | 4 | 0.40 |
| 2010 | Lefkosa | 0.20 | 5 | 0.80 |
| 2010 | Famagusta | 0.80 | 1 | 0.20 |
| 2010 | Kyrenia | 0.80 | 1 | 0.20 |
| 2010 | Guzelyurt | 0.75 | 2 | 0.25 |
| 2010 | Iskele | 0.70 | 3 | 0.30 |
| 2011 | Lefkosa | 0.45 | 3 | 0.55 |
| 2011 | Famagusta | 0.95 | 1 | 0.05 |
| 2011 | Kyrenia | 0.86 | 2 | 0.14 |
| 2011 | Guzelyurt | 0.50 | 4 | 0.50 |
| 2011 | Iskele | 0.48 | 5 | 0.52 |
| 2012 | Lefkosa | 0.62 | 2 | 0.38 |
| 2012 | Famagusta | 0.68 | 1 | 0.32 |
| 2012 | Kyrenia | 0.61 | 3 | 0.39 |
| 2012 | Guzelyurt | 0.40 | 5 | 0.60 |
| 2012 | Iskele | 0.58 | 4 | 0.44 |
| 2013 | Lefkosa | 0.34 | 3 | 0.66 |
| 2013 | Famagusta | 0.58 | 1 | 0.42 |
| 2013 | Kyrenia | 0.28 | 4 | 0.72 |
| 2013 | Guzelyurt | 0.38 | 2 | 0.62 |
| 2013 | Iskele | 0.58 | 1 | 0.42 |
| 2014 | Lefkosa | 0.61 | 4 | 0.39 |
| 2014 | Famagusta | 0.84 | 2 | 0.16 |
| 2014 | Kyrenia | 0.93 | 1 | 0.07 |

**Table 7.** *Cont.*

| Year | DMUs | Technical Efficiency Scores | Technical Rank | Technical Inefficiency Sores |
|------|------|------|------|------|
| 2014 | Guzelyurt | 0.72 | 3 | 0.28 |
| 2014 | Iskele | 0.28 | 5 | 0.72 |
| 2015 | Lefkosa | 0.57 | 4 | 0.43 |
| 2015 | Famagusta | 1 | 1 | 0 |
| 2015 | Kyrenia | 0.78 | 2 | 0.22 |
| 2015 | Guzelyurt | 0.53 | 5 | 0.47 |
| 2015 | Iskele | 0.30 | 3 | 0.70 |
| 2016 | Lefkosa | 0.88 | 3 | 0.12 |
| 2016 | Famagusta | 0.91 | 1 | 0.09 |
| 2016 | Kyrenia | 0.90 | 2 | 0.10 |
| 2016 | Guzelyurt | 0.91 | 1 | 0.09 |
| 2016 | Iskele | 0.51 | 4 | 0.49 |
| 2017 | Lefkosa | 0.80 | 5 | 0.20 |
| 2017 | Famagusta | 0.80 | 1 | 0.20 |
| 2017 | Kyrenia | 0.78 | 1 | 0.22 |
| 2017 | Guzelyurt | 0.75 | 2 | 0.25 |
| 2017 | Iskele | 0.65 | 3 | 0.35 |
| 2018 | Lefkosa | 0.84 | 3 | 0.16 |
| 2018 | Famagusta | 0.86 | 1 | 0.14 |
| 2018 | Kyrenia | 0.95 | 2 | 0.05 |
| 2018 | Guzelyurt | 0.48 | 4 | 0.52 |
| 2018 | Iskele | 0.50 | 5 | 0.50 |

Input Items: (1) WSEXP, (2) OEXP, (3) TREXP, (4) CAPEXP
Output Items: (1) POP, (2) NOLP, (3) NOWC, (4) HWCW

**Table 8.** Malmquist results.

Input Items: (1) WSEXP, (2) OEXP, (3) TREXP, (4) CAPEXP
Output items: (1) POP, (2) NOLP, (3) NOWC, (4) HWCW

| Year | DMUs | Technical Efficiency Scores | Tech Rank | TFP | Prod. Rank |
|------|------|------|------|------|------|
| 2004 | Lefkosa | 0.80 | 1 | 0.85 | 1 |
| 2004 | Famagusta | 0.76 | 2 | 0.78 | 2 |
| 2004 | Kyrenia | 0.62 | 4 | 0.64 | 4 |
| 2004 | Guzelyurt | 0.65 | 3 | 0.66 | 3 |
| 2004 | Iskele | 0.60 | 5 | 0.61 | 5 |
| 2005 | Lefkosa | 0.79 | 2 | 0.80 | 2 |
| 2005 | Famagusta | 0.80 | 1 | 0.83 | 1 |
| 2005 | Kyrenia | 0.63 | 4 | 0.65 | 4 |
| 2005 | Guzelyurt | 0.66 | 3 | 0.70 | 3 |
| 2005 | Iskele | 0.61 | 5 | 0.62 | 5 |
| 2006 | Lefkosa | 0.86 | 3 | 0.83 | 2 |
| 2006 | Famagusta | 0.79 | 4 | 0.80 | 3 |
| 2006 | Kyrenia | 0.90 | 1 | 0.88 | 1 |
| 2006 | Guzelyurt | 0.82 | 2 | 0.80 | 3 |
| 2006 | Iskele | 0.50 | 5 | 0.74 | 4 |
| 2007 | Lefkosa | 0.88 | 2 | 0.87 | 3 |
| 2007 | Famagusta | 0.91 | 1 | 0.92 | 1 |
| 2007 | Kyrenia | 0.87 | 3 | 0.88 | 2 |
| 2007 | Guzelyurt | 0.82 | 4 | 0.83 | 4 |
| 2007 | Iskele | 0.68 | 5 | 0.69 | 5 |
| 2008 | Lefkosa | 0.79 | 3 | 0.80 | 3 |
| 2008 | Famagusta | 1 | 1 | 1 | 1 |
| 2008 | Kyrenia | 0.76 | 4 | 0.77 | 4 |

**Table 8.** *Cont.*

| | Input Items: (1) WSEXP, (2) OEXP, (3) TREXP, (4) CAPEXP Output items: (1) POP, (2) NOLP, (3) NOWC, (4) HWCW | | | | |
|---|---|---|---|---|---|
| Year | DMUs | Technical Efficiency Scores | Tech Rank | TFP | Prod. Rank |
| 2008 | Guzelyurt | 0.86 | 2 | 0.85 | 2 |
| 2008 | Iskele | 0.63 | 5 | 0.70 | 5 |
| 2009 | Lefkosa | 0.85 | 4 | 0.81 | 3 |
| 2009 | Famagusta | 0.89 | 2 | 0.90 | 1 |
| 2009 | Kyrenia | 0.62 | 5 | 0.65 | 4 |
| 2009 | Guzelyurt | 0.99 | 1 | 0.90 | 1 |
| 2009 | Iskele | 0.87 | 3 | 0.88 | 2 |
| 2010 | Lefkosa | 0.30 | 5 | 0.33 | 5 |
| 2010 | Famagusta | 1 | 1 | 0.98 | 1 |
| 2010 | Kyrenia | 0.85 | 2 | 0.88 | 2 |
| 2010 | Guzelyurt | 0.78 | 4 | 0.74 | 4 |
| 2010 | Iskele | 0.79 | 3 | 0.75 | 3 |
| 2011 | Lefkosa | 0.29 | 5 | 0.28 | 4 |
| 2011 | Famagusta | 0.94 | 1 | 0.95 | 1 |
| 2011 | Kyrenia | 0.70 | 3 | 0.71 | 3 |
| 2011 | Guzelyurt | 0.83 | 2 | 0.84 | 2 |
| 2011 | Iskele | 0.20 | 4 | 0.25 | 5 |
| 2012 | Lefkosa | 0.80 | 1 | 0.83 | 1 |
| 2012 | Famagusta | 0.79 | 2 | 0.80 | 2 |
| 2012 | Kyrenia | 0.61 | 5 | 0.62 | 5 |
| 2012 | Guzelyurt | 0.66 | 3 | 0.70 | 3 |
| 2012 | Iskele | 0.63 | 4 | 0.64 | 4 |
| 2013 | Lefkosa | 0.86 | 2 | 0.83 | 2 |
| 2013 | Famagusta | 0.82 | 3 | 0.80 | 3 |
| 2013 | Kyrenia | 0.90 | 1 | 0.88 | 1 |
| 2013 | Guzelyurt | 0.79 | 4 | 0.80 | 3 |
| 2013 | Iskele | 0.50 | 5 | 0.74 | 4 |
| 2014 | Lefkosa | 0.86 | 2 | 0.83 | 3 |
| 2014 | Famagusta | 0.88 | 1 | 0.91 | 1 |
| 2014 | Kyrenia | 0.85 | 3 | 0.88 | 2 |
| 2014 | Guzelyurt | 0.80 | 4 | 0.83 | 4 |
| 2014 | Iskele | 0.66 | 5 | 0.70 | 5 |
| 2015 | Lefkosa | 0.81 | 4 | 0.81 | 3 |
| 2015 | Famagusta | 0.85 | 2 | 0.88 | 1 |
| 2015 | Kyrenia | 0.58 | 5 | 0.59 | 4 |
| 2015 | Guzelyurt | 0.95 | 1 | 0.98 | 1 |
| 2015 | Iskele | 0.83 | 3 | 0.84 | 2 |
| 2016 | Lefkosa | 0.92 | 2 | 0.93 | 2 |
| 2016 | Famagusta | 1 | 1 | 0.99 | 1 |
| 2016 | Kyrenia | 0.91 | 3 | 0.92 | 3 |
| 2016 | Guzelyurt | 0.88 | 5 | 0.87 | 5 |
| 2016 | Iskele | 0.89 | 4 | 0.89 | 4 |
| 2017 | Lefkosa | 0.90 | 1 | 0.88 | 1 |
| 2017 | Famagusta | 0.79 | 4 | 0.80 | 4 |
| 2017 | Kyrenia | 0.80 | 3 | 0.83 | 2 |
| 2017 | Guzelyurt | 0.82 | 2 | 0.81 | 3 |
| 2017 | Iskele | 0.66 | 5 | 0.70 | 5 |
| 2018 | Lefkosa | 0.80 | 1 | 0.83 | 1 |
| 2018 | Famagusta | 0.61 | 2 | 0.62 | 5 |
| 2018 | Kyrenia | 0.79 | 2 | 0.80 | 2 |
| 2018 | Guzelyurt | 0.66 | 3 | 0.70 | 3 |
| 2018 | Iskele | 0.63 | 4 | 0.64 | 4 |

**Table 9.** Average results for DEA and Malmquist Index.

| Municipality | DEA Average Technical Score | Malmquist Index Average Productivity Score |
|---|---|---|
| Lefkosa | 0.61 | 0.77 |
| Famagusta | 0.82 | 0.86 |
| Kyrenia | 0.70 | 0.76 |
| Guzelyurt | 0.62 | 0.80 |
| Iskele | 0.57 | 0.65 |
| Total | 0.66 | 0.77 |

Table 9 presents the average technical efficiency and productivity scores of both the DEA and the Malmquist index for each municipality and their total average scores.

Table 10 shows the relationship between the population size of the cities under question and the efficiency of city governments. According to the literature, we expect to reach a result that represents a positive relationship between population and efficiency score. The results show that this argument is met in all municipalities except for Nicosia, where the lowest efficiency score exists with the highest population. This can be explained by the employee size of the government that is an important determinant for the efficiency score. The number of employees in Nicosia is nearly double that of the other municipalities. Therefore, Nicosia has the lowest percentage of the number of people per staff. The general argument in the literature reveals that, if the number of employees in a local government increases, the efficiency of the city government decreases. The evidence supports this argument, in that Famagusta, which has the lowest percentage of the number of people per staff, is more efficient than the other municipalities.

**Table 10.** Analysis of Population Size and Efficiency of Government (2018).

| Municipality | Population | Technical Efficiency | Productivity |
|---|---|---|---|
| Lefkosa | 56,146 | 0.45 | 0.28 |
| Famagusta | 35,785 | 0.86 | 0.95 |
| Kyrenia | 27,357 | 0.85 | 0.71 |
| Guzelyurt | 18,562 | 0.50 | 0.84 |
| Iskele | 7222 | 0.48 | 0.25 |

Table 11 shows the relationship between the expenditure level, independent revenue source of local governments (grants), and the efficiency of municipalities. The general argument refers to the fact that, as an expenditure of a municipality increases, the efficiency of the municipality decreases. Table 11 provides proof for the general argument; however, the efficiency score of Iskele disputes this argument because Iskele has the lowest expenditure and the lowest efficiency score. This result is probably due to the reason explained in Table 8: except for Iskele, the North Cypriot municipalities support the general argument. The low efficiency score of Iskele municipality may be accepted given that the results do not follow a sustainable business model or may not reach enough resources to provide an efficient services.

**Table 11.** Expenditure, Independent Revenue Sources (Grants) and Efficiency of Municipalities between 2004 and 2018.

| Municipality | Average Total Expenditure | Average Total Grants | Technical Efficiency |
|---|---|---|---|
| Lefkosa | 59,472,449 | 31,412,793 | 0.45 |
| Famagusta | 28,105,036 | 12,713,795 | 0.86 |
| Kyrenia | 24,030,190 | 107,81,217 | 0.85 |
| Guzelyurt | 10,599,765 | 6,658,631 | 0.50 |
| Iskele | 7,204,029 | 1,885,514 | 0.48 |



The general argument states that, if a municipality depends on outside sources, this will lead to more inefficient operating activities. Therefore, when we analyzed Table 9, we realized that this argument is also relevant for the Cypriot municipalities. We can conclude that, if a municipality has more independent revenue sources, it is likely to be a more efficient municipality.

Before the Tobit analysis, the correlation coefficients in inflows and outflows are calculated, and we have decided there is no need to make any changes with the inputs and outputs.

Table 12 represents the Tobit results in the year 2004, which refer to salary expenses, the number of people belonging to each municipalities, total revenue from drinking water, total transfers that lead to decreases in the total efficiency, and productivity. However, transfers and population that have a positive signs lead to decreases in the total productivity. Therefore, more attention must be given to these insignificant variables. Table 13 represents the Tobit results in the year 2018 which show that, during this period, municipalities must give more attention to all variables except the total salary expenses, which is significant.

**Table 12.** Tobit Results 2004.

| Technicale~s | Coef. | Std.Err. | T | P > |t| | [95% Conf.Interval] | |
|---|---|---|---|---|---|---|
| Population | 0.0000443 | 0.0000156 | 2.84 | 0.025 | $7.47 \times 10^6$ | 0.0000811 |
| Nolp | $-0.0025883$ | 0.00511175 | $-0.51$ | 0.629 | $-0.0146892$ | 0.0095127 |
| Tow | 0.001293 | 0.0044785 | 0.29 | 0.781 | $-0.009297$ | 0.0118829 |
| Water | $-0.0016304$ | 0.0005582 | $-2.92$ | 0.022 | $-0.0029504$ | $-0.0003104$ |
| Sal exp | $-1.47 \times 10^{-7}$ | $6.02 \times 10^8$ | $-2.44$ | 0.045 | $-2.89 \times 10^7$ | $-4.38 \times 10^9$ |
| Operatingexp | $-6.31 \times 10^{-8}$ | $5.70 \times 10^8$ | $-1.11$ | 0.305 | $-1.98 \times 10^7$ | $7.18 \times 10^8$ |
| Transfers | $5.75 \times 10^7$ | $3.15 \times 10^7$ | 1.83 | 0.110 | $-1.69 \times 10^7$ | $1.32 \times 10^6$ |
| Capexp | $1.47 \times 10^8$ | $3.12 \times 10^8$ | 0.47 | 0.651 | $-5.90 \times 10^8$ | $8.84 \times 10^8$ |
| Cons | 0.4130972 | 0.0799435 | 5.17 | 0.001 | 0.2240609 | 0.6021335 |
| /sigma | 0.0854726 | 0.0164818 | | | 0.0464994 | 0.1244457 |

**Table 13.** Tobit Results 2018.

| Technicale~s | Coef. | Std.Err. | T | P > |t| | [95% Conf.Interval] | |
|---|---|---|---|---|---|---|
| Population | 0.0006609 | 0.0015444 | $-0.43$ | 0.682 | $-0.0043129$ | 0.0029911 |
| Nolp | $-0.0007144$ | 0.0055964 | $-0.13$ | 0.902 | $-0.0139477$ | 0.0125188 |
| Tow | 0.0001889 | 0.0018754 | 0.10 | 0.923 | $-0.0042458$ | 0.0046235 |
| Water | $-0.0003364$ | 0.0003664 | $-0.92$ | 0.389 | $-0.0012029$ | 0.00053 |
| Sal exp | $6.62 \times 10^{-8}$ | $1.84 \times 10^8$ | 3.59 | 0.009 | $-2.25 \times 10^8$ | $1.10 \times 10^7$ |
| Operatingexp | $6.76 \times 10^{-9}$ | $4.17 \times 10^8$ | 0.16 | 0.876 | $-9.17 \times 10^8$ | $1.05 \times 10^7$ |
| Transfers | $3.59 \times 10^8$ | $4.27 \times 10^8$ | 0.84 | 0.429 | $-6.52 \times 10^8$ | $1.37 \times 10^7$ |
| Capexp | $-7.47 \times 10^8$ | $4.39 \times 10^8$ | $-1.70$ | 0.132 | $-1.78 \times 10^7$ | $2.90 \times 10^8$ |
| Cons | 0.7455721 | 0.2821779 | 2.64 | 0.033 | 0.0782823 | 1.412772 |
| /sigma | 0.0833122 | 0.0159386 | | | 0.0456235 | 0.1210009 |

Table 14 represents that the total population has a significant and positive impact on the total productivity. This means that population cause an increase in technical efficiency. In addition, salary expenses, operating expenses, and capital expenses have a sign as expected, significant and negative signs. So, any decreases in the expenditures lead to increases in the productivity.

**Table 14.** Tobit Results between 2004–2018.

| Technicale~s | Coef. | Std.Err. | T | P > |t| | [95% Conf.Interval] | |
|---|---|---|---|---|---|---|
| Population | 0.0000159 | $3.69 \times 10^6$ | 4.31 | 0.000 | $8.42 \times 10^6$ | 0.0000235 |
| Nolp | 0.0033751 | 0.0021087 | 1.60 | 0.119 | $-0.0009201$ | 0.0076703 |
| Tow | 0.0008619 | 0.0007333 | 1.18 | 0.248 | $-0.0006317$ | 0.0023555 |
| Water | $-0.0002501$ | 0.0001905 | $-1.31$ | 0.198 | $-0.0006381$ | $-0.0001378$ |
| Sal exp | $-2.85 \times 10^{-8}$ | $9.07 \times 10^9$ | $-3.14$ | 0.004 | $-4.70 \times 10^8$ | $-1.0 \times 10^8$ |
| Operatingexp | $-1.01 \times 10^{-7}$ | $2.64 \times 10^8$ | $-3.82$ | 0.001 | $-1.55 \times 10^7$ | $-4.71 \times 10^8$ |
| Transfers | $-3.43 \times 10^8$ | $1.70 \times 10^8$ | $-2.01$ | 0.053 | $-6.89 \times 10^8$ | $3.94 \times 10^{10}$ |
| Capexp | $2.80 \times 10^8$ | $1.55 \times 10^8$ | 1.80 | 0.081 | $-3.61 \times 10^9$ | $5.96 \times 10^8$ |
| Cons | 0.5829133 | 0.0643611 | 9.06 | 0.000 | 0.451814 | 0.7140126 |
| /sigma | 0.1516308 | 0.0172977 | | | 0.1163966 | 0.1868649 |

The Tobit results suggest that total population has a positive impact, and that total salary expenses, operating expenses, and capital expenses have a negative impact on the total productivity. However, the results of DEA analysis in this study produce a contradiction to the literature, suggesting that the highly developed countries are more efficient in implementing the policy of sustainable development in the ecological dimension. However, the capital city of Nicosia in this study is accepted as the least efficient municipalities.

As mentioned in the study of Balaguer-Coll and Prior (2009) and Sole-Olle and Sorribas-Navarro (2011), grant has a significant effect on the efficiency, and our results indicate that lower expenditures and grants result in a higher efficiency. Therefore, we can conclude that municipalities, which have lower expenditures and grants, are more likely to function efficiently.

## 4. Conclusions

According to these results it can be concluded that municipalities in Famagusta and Kyrenia operate in the most efficient manner, in terms of technical efficiency and productivity. As it was explained, the results of the efficiency scores can be explained by different factors, such as economic, financial and environmental.

First, the inefficiency score can be explained through population size. The size of the economically active population of a city has a positive effect on the city's governments. However, this assumption cannot be accepted in the Nicosia situation. Consequently, this situation can gain clarity when the percentage of people per staff is taken into account. Therefore, we can conclude that the number of employees in the municipalities has a negative effect on the efficiency score. While evaluating the effect of the population in increasing productivity, it should not be forgotten that it may also help to deteriorate ecological sustainability. This might be controlled with implementing a sustainable business model.

The second conclusion is based on the ideas that lower expenditures and grants result in a higher efficiency. From this idea, independent revenue sources (grants) and the per capita expenditures of North Cypriot municipalities have a negative effect on the efficiency. Increases in grants leads to decreases in the efficiency, as it is apparent in the Nicosia municipality. In addition to this, other effects occur with increases in the total expenditures. Therefore, we can conclude that municipalities, which have lower expenditures and grants, are more likely to function efficiently.

In addition, the Tobit results represent that total population has a positive impact and that total salary expenses, operating expenses, and capital expenses have a negative impact on the total productivity. Therefore, we expect that any increases in the population leads to increases in the total productivity, decreases in the expenditures lead to increases in the productivity, and increases in the total transfers may lead to decreases in the total productivity. This paper implements a data envelopment analysis (DEA) to determine the efficiency of local governments and the factors that are important for the implementation of the sustainable development policy for decision makers. Therefore, the results help us to determine a way to provide efficient services by inefficient objects. The results may be

considered by local governments on the implementation of a sustainable development policy for their work.

Sustainable development will be possible with the mission, vision, and strategic goals of municipalities to direct and shape all the activities of the institution and to determine the performance indicators that affect the successes.

In order to progress sustainability as a reform in local government, it is highly advised that local government managers focus on integrating sustainability into strategic planning processes. Local government professionals, public management scholars, and policy makers should be urged to develop partnerships on decision-making in local government in order to investigate the hypotheses related to sustainability management. It should be noted that municipalities that have the same duties and responsibilities can reach a better situation by developing a common success criteria, as well as by managers discussing and exchanging ideas the status, priorities, problems, and solutions of their institutions. In addition, as it is mentioned in the study of Ziemba (2021), ICT quality, information culture, and ICT management might have a significant positive impact on the sustainability of local governments. Therefore, further research should consider the importance of ICT on the level of efficiency in North Cyprus municipalities.

Within this study, using proper waste mechanism, clean water technology and solar system for lighting are the indicators that frugal innovation can be applicable for the local governments to preserve the environment. Further research should compare the results with the southern part of a country to the analysis on the efficiency of the implementation of the sustainable development policy in both the economic and socio-cultural dimensions.

We evaluated the efficiency of the municipalities on the island as part of this study, which shed light on the state authorities. We expose the causes that will help state officials to improve municipal management by examining the efficiency of the five main district municipalities mentioned in this study. The fact that such a study has never been carried out in this country before is extremely significant for the country's municipalities.

**Funding:** This research received no external funding.

**Institutional Review Board Statement:** Not applicable.

**Informed Consent Statement:** Not applicable.

**Data Availability Statement:** The data used in the study was obtained from the Ministry of Interior and the accounting departments of the municipalities. Data access has been taken with the agreement of the author and the ministry, and restrictions have been imposed on its presentation to everyone. Data will be made available upon request.

**Acknowledgments:** I must acknowledge the staff at the interior ministry of Northern Cyprus for their co-operation in providing data and materials for this thesis.

**Conflicts of Interest:** The authors declare no conflict of interest.

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
