# Peer review of "The Turkish Cypriot Municipalities’ Productivity and Performance: An Application of Data Envelopment Analysis and the Tobit Model"

_jrfm, doi:10.3390/jrfm14090407_

Round 1
Reviewer 1 Report
Thank you for inviting me as a reviewer for the manuscript titled The Turkish Cypriot Municipalities’ Productivity and Performance in the Perspective of Frugal Innovation: An Application of Data Envelopment Analysis and Tobit Model. For the final acceptance of the paper, the following corrections must be made:
- Need to better highlight the novelty of the study in the introduction. Add information about the existing scientific gap and the reason why the issue is worthy to solve.
- Literature analysis needs to be improved. In the entire, there is only four paper from 2019-2021. Add another 10-15 papers of more recent date (period 2019-2021), such as:
- Biswas, S., Bandyopadhyay, G., Guha, B., & Bhattacharjee, M. (2019). An ensemble approach for portfolio selection in a multi-criteria decision making framework. Decision Making: Applications in Management and Engineering, 2(2), 138-158. https://doi.org/10.31181/dmame2003079b.
- Hassanpour, M. (2020). Evaluation of Iranian small and medium-sized industries using the deabased on additive ratio model–A review, Facta Universitatis Series: Mechanical Engineering, 18(3), 2020, 491-511. DOI: https://doi.org/10.22190/FUME200426030H.
- Show a sensitivity analysis of the model.
- Show in detail the advantages and limitations of the proposed methodology and this study.
Author Response
Dear Reviewer;
In response to your suggestions, I've updated my work. The changes are shown in red.
Best Regards

Reviewer 2 Report
The paper deals with the study on „The Turkish Cypriot Municipalities’ Productivity and Performance in the Perspective of Frugal Innovation: An Application of Data Envelopment Analysis and Tobit Model”. The presented topic is of high professional and practical interest what brings a significant added value to potential target group of readers. The overall writing style reflects a logical clear concept. Hopefully, my remarks, observations, and possible suggestions might bring the authors benefits for the enhancement of the paper to be published properly. Accordingly, I am stating my comments below.
Title
The title reflects the objective and content of the paper, the lenght is too long. Please reduce the lenght.
Abstract
The abstract provides a structured summary including contextual background, and result, conclusion, and implications of key findings, etc.
Introduction
This part of the paper is properly designed in a correct explanatory way. This part does highlight the aims of this investigation. The most relevant part is the introduction section that gives a perfect context for the justification of the research.
The Introduction section is practically combined with the literature review. This section includes many relevant references and authors provide solid theoretical foundations for the analysis using appropriate references.
Methodology
Methodology explained and described in a clear way. Data about quantitative and comparative analyses are provided in the manuscript. The methodology is suitable for the research objectives. The theoretical model and the hypotheses were well established and supported by the literature.
Results
The results that are reported refer to figures that are known in detail both in the text and in the illustrations. Obtained results are developed based on methodological justification, they are supported by relevant presentation. Clarity in tables and graphs is appropriate. The presentation of material is logical and technically correct. The tables presented here are justified for an adequate statement of results.
Conclusion
Conclusions are logically ordered.
The interpretations and conclusions are sound and justified by the results.
Author Response
Dear Reviewer;
In response to your and other reviewer suggestions, I've updated my work. The changes are shown in red.
Best Regards

Reviewer 3 Report
This paper empirically investigates the efficiency, technical efficiency, productivity, and their determinants factor of implementing the sustainable development policy of the five major municipalities in North Cyprus by conducting DEA and Tobit analyses during the period of 2004 to 2018 quarterly.
The article is interesting and valuable for science. The argumentation of the article is confirmed by the necessary calculations.
The article used advanced statistical and econometric methods were.
Research methods are correct and meaningful.
The results of the investigation are linked to the methodology. Every result is reasonable and describes the goal of a scientific paper.
Author Response
Dear Reviewer;
Thank you for your positive responsesabout my work
Best Regards
Round 2
Reviewer 1 Report
All the reviewers' comments have been addressed carefully and sufficiently. The revisions are rational from my point of view. I think the current version of the paper can be accepted.